# INSIDER: Interpretable sparse matrix decomposition for RNA expression data analysis

Kai Zhao[1], Sen Huang[2], Cuichan Lin[3], Pak Chung Sham[4,5,6], Hon-Cheong So[3,7,8]*, Zhixiang Lin[1]*

1 Department of Statistics, The Chinese University of Hong Kong, Shatin, Hong Kong SAR, China, 2 Department of System Engineering and Engineering Management, The Chinese University of Hong Kong, Shatin, Hong Kong SAR, China, 3 Department of Psychiatry, The Chinese University of Hong Kong, Shatin, Hong Kong SAR, China, 4 Department of Psychiatry, University of Hong Kong, Pokfulam, Hong Kong, China, 5 Centre for Genomic Sciences, University of Hong Kong, Pokfulam, Hong Kong, China, 6 State Key Laboratory for Cognitive and Brain Sciences, University of Hong Kong, Pokfulam, Hong Kong, China, 7 School of Biomedical Sciences, The Chinese University of Hong Kong, Shatin, Hong Kong SAR, China, 8 KIZ-CUHK Joint Laboratory of Bioresources and Molecular Research of Common Diseases, Kunming Institute of Zoology and The Chinese University of Hong Kong, Hong Kong, China

* hcso@cuhk.edu.hk (H-CS); zhixianglin@cuhk.edu.hk (ZL)

**Data Availability Statement:** o The BrainSpan data can be freely downloaded at its official website (https://www.brainspan.org/static/download.html), which includes normalized expression values and meta-data. The version of RNA-Seq Gencode v10

## Abstract

RNA sequencing (RNA-Seq) is widely used to capture transcriptome dynamics across tissues, biological entities, and conditions. Currently, few or no methods can handle multiple biological variables (e.g., tissues/ phenotypes) and their interactions simultaneously, while also achieving dimension reduction (DR).

We propose INSIDER, a general and flexible statistical framework based on matrix factorization, which is freely available at https://github.com/kai0511/insider. INSIDER decomposes variation from different biological variables and their interactions into a shared low-rank latent space. Particularly, it introduces the elastic net penalty to induce sparsity while considering the grouping effects of genes. It can achieve DR of high-dimensional data (of >= 3 dimensions), as opposed to conventional methods (e.g., PCA/NMF) which generally only handle 2D data (e.g., sample × expression). Besides, it enables computing 'adjusted' expression profiles for specific biological variables while controlling variation from other variables. INSIDER is computationally efficient and accommodates missing data. INSIDER also performed similarly or outperformed a close competing method, SDA, as shown in simulations and can handle complex missing data in RNA-Seq data. Moreover, unlike SDA, it can be used when the data cannot be structured into a tensor. Lastly, we demonstrate its usefulness via real data analysis, including clustering donors for disease subtyping, revealing neuro-development trajectory using the BrainSpan data, and uncovering biological processes contributing to variables of interest (e.g., disease status and tissue) and their interactions.

summarized to genes is used in our work. The RNA sequencing data and meta-data for the Aging, Dementia and Traumatic Brain Injury Study of 107 brains, including 377 samples from cortical grey (parietal and temporal), white matter (parietal) and hippocampus), is freely available at its official website (http://aging.brain-map.org/download/index) or via https://www.ncbi.nlm.nih.gov/geo/query/acc.cgi?acc=GSE104687. The extracted GTEx (Genotype-Tissue Expression) data is obtained from the RNA sequencing data offered by the GTEx project. The RNA-Sequencing dataset is open access at its official portal (https://gtexportal.org/home/downloads/adult-gtex/overview). The software for INSIDER is freely available on GitHub (https://github.com/kai0511/insider).

**Funding:** ZL has been supported by the Chinese University of Hong Kong startup grant (4930181), the Chinese University of Hong Kong Science Faculty's Collaborative Research Impact Matching Scheme (CRIMS 4620033), the Chinese University of Hong Kong direct grants (4053540, 4053586), and Hong Kong Research Grant Council (GRF 14301120, 14300923). The funders play no role in the study design, data collection and analysis, decision to publish, or preparation of the manuscript.

**Competing interests:** The authors have declared that no competing interests exist.

## Author summary

The RNA-Seq studies aim to understand the contributing role of gene activities to phenotypes, especially human disorders, reveal important biological processes (BPs) that contribute to phenotypes, and understand the involvement of tissues (e.g., brain structures/regions) in disease development. However, our ability to understand these key issues in high-dimensional settings is hampered by the limitations of currently available methods for analyzing RNA-seq data. To address these issues, we propose a novel statistical method to jointly model variation from multiple biological variables (e.g., donor, tissue, phenotype), to introduce interactions (tissue × phenotype), and to reduce dimensionality. The favorable properties of our method are discussed below. Its advantages over one closest existing computational tool and its wide applicability are shown via simulation and applications. A computationally efficient implementation of our method is provided.

## Introduction

RNA Sequencing (RNA-Seq), which has been developed for more than a decade, uses deep sequencing technology to offer a far more precise measurement of levels of transcripts and their isoforms than other methods [1]. It is now widely used to measure transcriptome levels across different tissues from a number of donors and capture transcriptome dynamics across tissues or biological conditions. For example, the Genotype-Tissue Expression (GTEx) [2] project offers a valuable data resource to study tissue-specific gene expression and regulation. Moreover, there are many data resources related to brain development or psychiatric disorders across brain regions, including the BrainSpan [3], PsychENCODE [4], and ATP (the Autism Tissue Program) [5]. The primary task of these studies is to link gene activities to phenotypes, especially human disorders, reveal important biological processes (BPs) that contribute to phenotypes, and understand the involvement of tissues (e.g., brain structures/regions) in disease development.

However, variation in transcript levels due to covariates, such as donor and tissue, makes the task challenging. Let us take studies of autism spectrum disorder (ASD) as examples. Different brain regions may demonstrate wide differences in gene activities in ASD [6]. On the other hand, no individuals with ASD have exactly the same genetic expression profiles [7], which are influenced by individual characteristics such as age, sex, diet, and lifestyle. Thus, joint analyses of gene expression by RNA-Seq of multiple brain regions from several donors are hindered by the variation from donors and tissues.

Tools for differential gene expression (DGE) analysis [8, 9], typically built upon generalized linear models (GLM), have been widely employed to quantify changes in expression levels across experimental groups or biological conditions in analyzing RNA expression data. Such methods can also consider confounding variation from covariates, such as donor, and tissue. However, DGE analysis does not aim for dimension reduction and thus cannot help achieve the tasks that require reducing high dimensional data onto a low-rank latent space, such as clustering donors based on low-rank representations of donors (low-rank representations are preferred as the high dimension of genomic features often makes clustering analysis difficult), and other downstream analysis with low-rank representations of biological conditions (phenotype, tissue, or developmental stage).

Traditional dimension reduction (DR) approaches, such as principal component analysis (PCA), factor analysis, and matrix factorization, cannot directly handle the variation from *multiple* covariates, such as donor, tissue, and other clinical or biological features, to facilitate

further analysis. One study [10] proposed a PCA-based approach with adjustment for confounding variables. Other approaches based on factor analysis have been proposed to adjust for technical artifacts or confounding variables [11,12]. These studies aim to remove technical artifacts or control confounding variations rather than interpret variations originating from biological conditions.

Non-negative matrix factorization (NMF) [13–15] has gained popularity in the analysis of genomic data. The conventional NMF method cannot be directly employed to model variations arising from *multiple* biological variables and their interactions, and to interpret variations from *multiple* biological features or conditions. Variants of NMF proposed for the analysis of bulk RNA-Seq data are rather limited, and most of them aim to correct batch effects in single-cell settings instead of interpreting variation from biological variables [14,15]. On the other hand, NMF and its variants fail to directly infer down-regulations of elements in biological settings [16], so its interpretability is hampered.

To summarize, conventional DR methods such as PCA and NMF can usually only be used for two-dimensional (2D) data (e.g., sample × expression), and cannot directly handle *multiple* variables or biological conditions (e.g., donor × tissue × expression; donor × time × expression etc.), or interactions between these conditions. On the other hand, the new methodology we proposed below (INSIDER) can handle *higher-dimensional data (of three or more dimensions)*, as well as interactions between the variables of interest, while performing DR simultaneously.

We note that most tensor decomposition approaches [17–19] can achieve DR with high-dimensional data, however most cannot handle missing data or invoke sparsity in the latent representations. This may impose limitations in analyzing gene expression data since the function of a particular biological pathway often only involves a subset of genes, and each latent dimension only encodes information related to a few pathways. Thus, a sparse Bayesian tensor decomposition approach [20] (Sparse Decomposition of Arrays [SDA]) was proposed to capture variation from donor and tissue in analyzing RNA-Seq data. However, Bayesian-based approaches face a heavy computational burden in analyzing large-scale genomic data, especially when optimizing with Markov Chain Monte Carlo.

To model variation from *multiple* biological variables and to facilitate interpretation of the variation, we propose a general and flexible statistical framework, INSIDER, based on matrix decomposition (MF) without non-negative constraints to decompose the variation from covariates in high dimensional data into a shared low-rank latent space. INSIDER has the following favorable properties:

1. it is a general and flexible framework, in which various covariates and interaction terms can be defined based on research questions and study design. Importantly, it can accommodate *multiple* biological variables simultaneously, and achieve DR of *high-dimensional* data, as opposed to other conventional methods such as PCA and NMF which generally can only handle 2D data (e.g., sample × expression);

2. it is able to consider *interactions* between different biological/clinical variables, enabling gaining insights into the effect of one variable on others. For example, modeling gene expression of various brain regions across neuropsychiatric disorders, the brain region × disorder interaction, can help understand how brain regions interact with the type of disorder to influence expression. Intuitively, the same disease may affect expression differently in different brain regions; such information may be missed if interactions are not modeled. Notably, interactions are usually ignored by conventional methods, such as PCA, NMF, and their variants. Interactions were also ignored in the recently developed SDA approach [20];

3. it allows one to compute the 'adjusted' expression for one variable (e.g., phenotype) that controls for variation from other variables (e.g., donor). In addition, to consider the group effect among genes and invoke sparsity in gene expression, it introduces an elastic net penalty (combination of L1 and L2 penalty). Elastic net penalty is widely used in the regression setting where only a subset of features may contribute to the outcome, but it prefers highly correlated predictors to be included or excluded from the model together [21] (instead of arbitrarily selecting one predictor over others as in LASSO);

4. Results from it facilitate different downstream analyses, including clustering donors with donor representations, temporal dynamic analyses, and uncovering disease mechanisms. The potential of INSIDER in facilitating clinical and biological discoveries will be demonstrated in the Results section;

5. it is computationally efficient and can also handle missing data well. These will be elaborated further in the sections below;

6. it also performed similarly or outperformed one of the closest competing methods, SDA, as shown in our simulations. Moreover, unlike SDA, it can also be used where the data matrix cannot be neatly structured into a tensor, as explained below.

To recap the uniqueness of INSIDER, it models variation arising from *multiple* biological variables and their interaction to comprehend their effects on gene expression in a complex biological setting instead of solely removing technical artifacts or controlling confounding variations. Moreover, it models the effect of specific biological variables while controlling for variation from other biological variables. Unlike GLM-based methods for DGE analysis, INSIDER is a statistical approach for dimension reduction, thus enabling performing the tasks that are challenging using high-dimensional features, such as clustering subjects with donor representations. In contrast to NMF, INSIDER considers the interaction between biological variables and does not impose non-negativity constraints on the latent representations, which allows for capturing down-regulation in gene expression and thus enhances model interpretability. Additionally, compared with tensor decomposition, it can handle situations where the data matrix cannot be structured into a tensor. Usually, for tensor decomposition approaches, we assume one observation/sample per each combination of the three (or more) dimensions, but this is not always the case in practice. To illustrate with an example, when analyzing the effect of neuropsychiatric disorders on expression in different brain regions, multiple samples may be taken from a specific brain region from donors with a certain disorder. The number of samples taken could also differ across donors. This scenario can be challenging for standard tensor decomposition or SDA [20], but can be handled by our proposed method.

## Results

### Overview of INSIDER

We propose a novel computational method called INSIDER (interpretable sparse matrix decomposition for bulk RNA expression data analysis), summarized in Fig 1. INSIDER considers that variation in RNA expression levels originates from several biological variables or covariates, such as donor, tissue, and phenotype (Fig 1A). The variation is decomposed into a shared latent space of rank $K$ by matrix factorization, and variation from unknow sources can also be considered by INSIDER (Fig 1B). The effect of each covariate on expression can be measured by multiplying the corresponding latent representation with the gene representation $V$ (Fig 1B). Specifically, the expression level of gene $m$ of sample $s$ from tissue $h$ of donor $i$ with

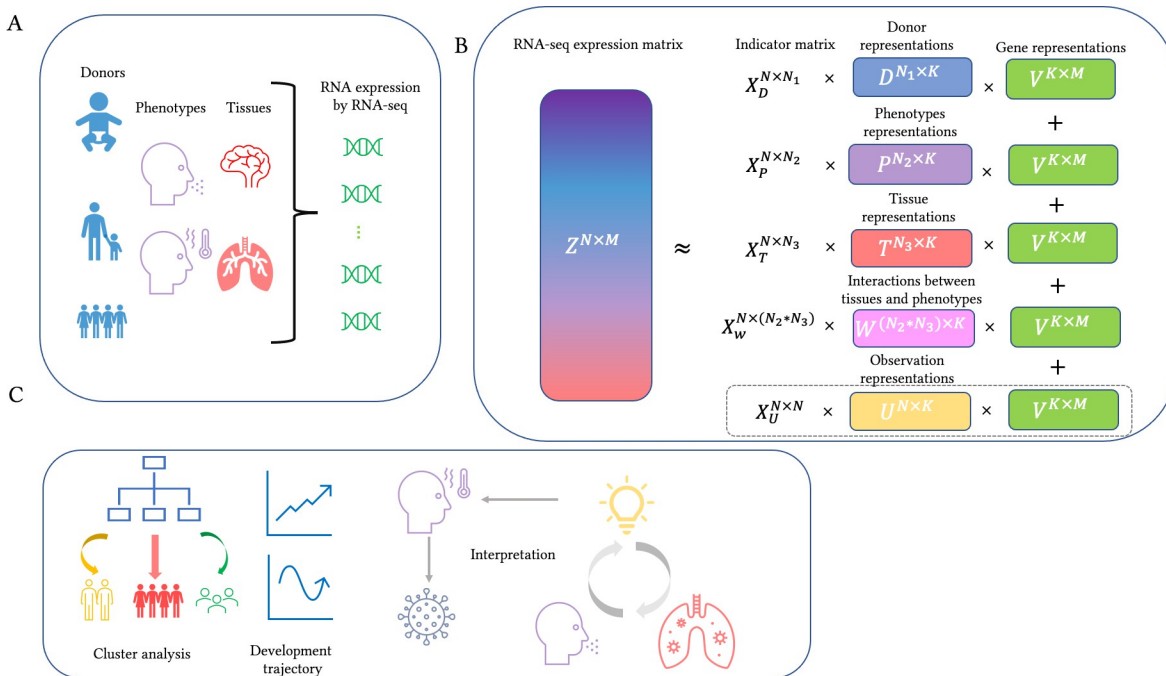

**Fig 1. Overview of INSIDER.** A. The RNA-seq data is usually obtained from samples of different tissues from several donors with different phenotypes. B. The variation in RNA-seq expression data can be decomposed into a shared *K*-rank latent space by INSIDER. The latent representation for observations *U* can be included when the donor, phenotype, and/or tissue information is unavailable. Note that INSIDER incorporates the interaction between covariates and that the gene representation *V* is shared. C. Various types of downstream analysis can be conducted on the result from INSIDER. The donor representations can be utilized for cluster analysis, development trajectories can be revealed in the analysis of the BrainSpan data, and biological mechanisms behind phenotypes of interest and interactions between covariates can be revealed. Details on these analyses will be demonstrated in the Result section.

phenotype *j*, $z_{ijhm}$, is modelled as

$$\hat{z}_{ijhm} = d_i^T v_m + p_j^T v_m + t_h^T v_m + \mathrm{w}_{jh}^T v_m + u_s^T v_m$$

where $d_i, p_j, t_h, w_{jh}, u_s, v_m$ are vectors of length *K*. $u_s$ is usually ignored when donor, tissue, and phenotype information is available and can be included to capture variation from unknown sources when part or all of the information is unavailable. The interaction effect between tissues and phenotypes is incorporated by introducing $w_{jh}$ on gene expression. For example, a certain disease may be associated with changes in gene expression, but the degree of change may differ by the tissue type. Let $Z^{N \times M}$ denote the observed data matrix for *N* samples and *M* genes. The above equation can be rewritten in the following matrix representation

$$\hat{Z} = (X_D D + X_P P + X_T T + X_W W + X_U U) V.$$

We assume there are $N_1$ donors, $N_2$ phenotypes, and $N_3$ tissues. For the purpose of representation, $X_D^{N \times N_1}, X_P^{N \times N_2}, X_T^{N \times N_3}, X_W^{N \times (N_2 * N_3)}, X_U^{N \times N}$ are introduced to be the indicator matrices for donors, phenotypes, tissues, phenotype by tissue interactions, and observations, which are the dummy design matrices for the samples. $X_U$ is an identity matrix. $D^{N_1 \times K}, P^{N_2 \times K}, T^{N_3 \times K}, W^{(N_2 * N_3) \times K}, U^{N \times K}$ are the matrices for latent representations for the effects of donors, phenotypes, tissues, phenotype by tissue interactions, and observations on the *K* latent metagenes, respectively, and $V^{K \times M}$ represents *K* metagenes across the *M* genes, which capture genes that tend to be co-expressed or work together to induce biological functions.

In INSIDER, we seek to minimize the following objective

$$\mathcal{L}(D, P, T, W, U, V) = \frac{1}{2} \| Z - (X_D D + X_P P + X_T T + X_W W + X_U U)V \|_F^2$$

$$+ \frac{1}{2} \lambda \left[ \| D \|_F^2 + \| P \|_F^2 + \| T \|_F^2 + \| W \|_F^2 + \| U \|_F^2 \right]$$

$$+ \lambda \left[ \frac{1}{2}(1 - \alpha) \| V \|_F^2 + \alpha |V|_1 \right].$$

Here $\lambda, \alpha$ are tuning parameters for the elastic net regularization term, $\|\cdot\|_F$ represents the Frobenius norm. Alternating block coordinate descent (BCD) is utilized to optimize the objective. INSIDER can incorporate observations with missing values. The technical details of INSIDER are presented in the Methods section.

INSIDER enables various downstream analyses (Fig 1C). For instance, cluster analysis on the donor representation can reveal characteristics of the donors and help find patients subgroups or disease subtyping, development trajectory can be more readily revealed with 'meta-genes' (low-rank representations of gene expression), which capture the information of genes that tend to be co-expressed or work together to induce biological functions [22]. Also, biological processes (BPs) enriched by results from INSIDER help to gain insights into mechanisms underlying covariates like phenotype and interactions between covariates. Details of these analyses are presented below.

## Simulation study

To our knowledge, there is no readily available computational tool for modeling the variation arising from *multiple* biological variables and their *interactions*, while also performing dimension reduction (and having similar favorable properties as described earlier). Tensor decomposition is the closest existing computational tool to INSIDER. To showcase the advantages of INSIDER in capturing complex latent structures of gene expression data, it was compared to SDA[20], a sparse Bayesian tensor decomposition approach. We evaluated their abilities to accurately recover the underlying latent structures from *multiple* biological variables and their *interactions* with RMSE (root-mean-squared error).

In our simulation setup, we consider a gene expression matrix $Z$ (250×200), and variation in the matrix comes from two biological variables ($E,F$), their interaction, and random noise. Their relationship is described by $Z = T + \Theta$, where $T = (X_E E + X_F F + X_R R)V$. The rank of latent space is set to 5, $E$ and $F$ are matrices of dimensions 50×5 and 5×5, respectively, and the dimension of $R$ for the interaction between $E$ and $F$ is 250×5. $V$ is a 5×200 matrix for gene latent representations. These matrices were generated from the standard normal distribution. The design matrices for $\{E,F,R\}$ are $X_E^{250 \times 50}$, $X_F^{250 \times 5}$, and $X_R^{250 \times 250}$, respectively. The noise matrix $\Theta$ (250×200) was generated from a normal distribution $N(0, \delta)$, and $\delta$ is the variance of the distribution. We considered $\delta$ equal to 0.25, 0.5, 0.75, and 1 in our simulations. Meanwhile, to introduce sparsity in gene expression, we randomly silence the expression of 30% of the genes.

Subsequently, we employed INSIDER with and without considering the interaction between $E$ and $F$ and SDA to decompose the matrix $Z$, and then evaluated their abilities to recover $T$ using RMSE. For the implementation of SDA, the software provided by the original study [20] was used. When running SDA, the parameters $N$ and *num_comps* are set to 50 and 5, respectively, and default values are used for other parameters. Results for the comparison are shown in Table 1.

**Table 1. Comparison of performance of INSIDER without or with interaction to that of SDA in recovering $T$ using RMSE when the data is generated according to INSIDER.**

| RMSE | noise variance $\delta$ | | | |
|---|---|---|---|---|
| | **0.25** | **0.5** | **0.75** | **1.0** |
| SDA | 2.21 | 2.12 | 1.82 | 1.84 |
| INSIDER without interaction | 1.69 | 1.69 | 1.69 | 1.69 |
| INSIDER with interaction | 0.05 | 0.10 | 0.16 | 0.21 |

From Table 1, we see that SDA fails to capture the underlying relationship across different noise levels. Moreover, INSIDER without interaction outperforms SDA in recovering $T$ using RMSE, even though both methods have a similar model complexity. Importantly, when considering the interactive effects between $E$ and $F$, INSIDER demonstrates substantially better performance in recovering $T$, compared with SDA, even as the level of noise increases.

To show the performance of INSIDER in case of model misspecification, we first included simulation studies of INSIDER under the situation where the simulation data is generated according to SDA [20], a tensor decomposition method. Specifically, we consider a tensor of gene expression $Z$ of dimensions 50×5×200, which is generated according to $Z = T + \Theta$, where $T_{nlg} = \sum_{c=1}^{K} E_{nc} F_{lc} V_{cg}$. Here the rank of latent space $K$ is set to 5, $E$ and $F$ are matrices of dimensions 50×5 and 5×5, respectively, and $V$ is a 5×200 matrix for gene latent representations. These matrices were generated from the standard normal distribution. The elements of $\Theta$ are generated from $N(0, \sigma)$. In practice, we consider $\sigma$ to be 0.25, 0.5, 0.75, and 1. For the application of INSIDER to the simulated data, we flat the tensor $Z$ by its first two dimensions and transform it into a 250×200 matrix. Then, we introduce two design matrices $X_E^{250×50}$ and $X_F^{250×5}$ for the first two dimensions of $Z$, and the two design matrices are considered as covariates for INSIDER. Since SDA only has one hyperparameter, the latent rank $K$ (*num_comps*), and does not allow missing entries, we adopt the following procedure in the simulation study. When running SDA, the parameters $N$ and *num_comps* are set to 50 and 5, respectively, and default values are used for other parameters. For INSIDER, we first draw 10% of the elements of $Z$ without replacement as validation set and consider the left data trainset, select the hyperparameters with which INSIDER trained on the trainset performs the best on the validation set, and run INSIDER with selected hyperparameters on $Z$. Then we compared the performance of INSIDER with and without interactions to SDA in recovering the true pattern $T$ using RMSE. The results of the simulation studies are presented in Table 2.

From Table 2, we find that INSIDER without interaction did not perform well in the case of model misspecification. However, INSIDER with interaction demonstrated a performance similar to SDA, although the simulation data is generated according to the SDA model. This suggests that INSIDER performs well in the case of model specification when data is generated according to SDA.

**Table 2. Comparison of the performance of INSIDER without or with interaction to that of SDA in recovering $T$ using RMSE when the data is generated according to SDA.**

| RMSE | noise variance $\delta$ | | | |
|---|---|---|---|---|
| | **0.25** | **0.5** | **0.75** | **1.0** |
| SDA | 0.04 | 0.08 | 0.11 | 0.16 |
| INSIDER without interaction | 1.22 | 1.57 | 2.10 | 1.66 |
| INSIDER with interaction | 0.05 | 0.11 | 0.16 | 0.21 |

To further demonstrate the performance of INSIDER in case of model misspecification, we conducted further simulation analysis to examine its performance when the data is generated according to multivariate linear regression models. For illustration, the gene expression matrix $Z$ of dimensions 100×200 is generated by $Z = T + \Theta$, where $T = X_E E + X_F F$, $X_E^{100 \times 20}$ and $X_F^{100 \times 5}$ are the dummy matrices for two categorical variables, $E^{20 \times 200}$, $F^{5 \times 200}$ are matrices for coefficients of the two categorical variables across 200 genes, and $\Theta^{100 \times 200}$ is the error matrix. In simulation, $E$ and $F$ are generated from $N(0, 1)$ and $\Theta$ is generated from $N(0, \sigma)$, where $\sigma$ denotes the noise level. We consider $\sigma$ to be 0.25, 0.5, 0.75, and 1. For INSIDER, we incorporate two categorical variables as covariates without considering their interaction. For linear regression, we fit ridge regression with the two categorical variables as predictors and $Z$ as responses. For a fair comparison, we first draw 10% of the elements of $Z$ without replacement as hold-out set, draw 10% of the elements of the left data without replacement as validation set, and consider other elements of the left data as trainset. To employ INSIDER and Ridge regression, we first train them on the trainset with different hyperparameters and select the parameters with which they perform the best on the validation set in terms of RMSE. Then, we fit them with the chosen hyperparameters on the left data (trainset + validation set) and report the RMSE of the fitted model on $T$ for the testset. The baseline RMSE is the root-mean-square difference between $T$ and the average of $Z$ over the trainset and validation set. The results of the simulation are shown in Table 3.

Table 3 shows that INSIDER performs similarly to ridge regression in terms of RMSE in recovering $T$. Moreover, compared with the baseline RMSE, INSIDER captures a large proportion of signals. This suggests that INSIDER performs well under the case of model specification when the simulated data is generated according to multivariate linear models.

Further, we also explored the ability of INSIDER to handle missing values in simulation settings. There are mainly two types of missing in RNA-Seq studies. The first type is missing samples. For example, the biosamples of some tissues from donors are not available, and this leads to missing samples in RNA-Seq data. The second type is missing entries. That is, expression levels of some genes for samples are missing. In practice, the proportion of missing entries is usually small in RNA-Seq data. Here we studied the two types of missing in the simulation below.

In our simulation study, we assume there are 50 donors with 2 different phenotypes, 8 tissues, and 1000 genes. The number of samples for our simulated data matrix is 400 (50×8). Moreover, we consider the interaction between phenotypes and tissues. In our simulation setup, we consider a gene expression matrix $Z$ (400×1000), and variation in the matrix comes from three biological variables $(D, P, T)$, the interaction between $P$ and $T$, and random noise. Their relationship is described by $Z = R + \Theta$, where $R = (X_D D + X_P P + X_T T + X_W W)V$. The rank of latent space is set to 5, $D$, $P$, and $T$ are matrices of dimensions 50×5, 2×5, and 8×5, respectively, and the dimension of $W$ for the interaction between $P$ and $T$ is 16×5. $V$ is a 5×1000 matrix for gene latent representations. All matrices for latent representation were generated from the standard normal distribution. The design matrices for $\{D, P, T, W\}$ are $X_D^{400 \times 50}$,

**Table 3. Performance of INSIDER in recovering $T$ over the hold-out set using RMSE when the data is generated according to a multivariate linear model.**

| RMSE | noise variance $\delta$ | | | |
|---|---|---|---|---|
| | **0.25** | **0.5** | **0.75** | **1.0** |
| Baseline | 1.4234 | 1.4046 | 1.4005 | 1.4001 |
| Ridge regression | 0.1496 | 0.2934 | 0.4257 | 0.5426 |
| INSIDER | 0.1493 | 0.2950 | 0.4268 | 0.5458 |

$X_P^{400\times2}$, $X_T^{400\times8}$, and $X_W^{250\times16}$, respectively. The noise matrix $\Theta$ (400×1000) was generated from a normal distribution $N(0, \delta)$, and $\delta$ is the variance of the distribution. In the simulation, we considered $\delta$ equal to 0.25, 0.5, 0.75, and 1.

To better mimic the complex missing samples in real settings, we consider the setting based on the distribution of brain structures and phenotypes of samples of 50 donors similar to the ageing, dementia, and TBI data. Among 50 donors, 27 are healthy, and the other 23 are affected by a specific disease. For each donor, at least 4 out of 8 tissues are available. Then, we had 229 samples available, and 171 (42.75%) samples were missing. For comparison, we also conducted simulation studies on the scenario where 42.75% of samples are missing at random. Further, to introduce missing entries in the data matrix of samples available, we randomly drew 5% of the elements from it as missing entries. To evaluate the performance of INSIDER in handling missing, we examined the RMSE between the estimated $\hat{R}$ by INSIDER and $R$ over missing entries and samples, and the result is presented in Table 4. In the table, the baseline RMSE is the root-mean-square difference between $R$ for missing entries or samples and the average of $Z$ over nonmissing entries. Since the baseline RMSE for missing entries and samples are roughly the same, we only show the baseline RMSE for missing samples for illustration.

From the table, we see that the RMSE for missing entries and samples increases slightly with increasing noise levels, and the RMSE for missing entries is slightly higher compared with the RMSE for missing samples. The RMSE for random missing samples and the designed missing samples based on the ageing, dementia, and TBI data is roughly the same. The simulation result suggests that the INSIDER has a good ability to recover underlying true patterns and the ability decreases slightly with increasing noise levels, even though complex missing exists.

## Analysis of the BrainSpan data

The BrainSpan study provides RNA-Seq data profiling up to sixteen cortical and subcortical structures across the entire course of human brain development [23]. In this analysis, we aim to explore the development trajectories across human brain development, the BPs that function in different brain regions, and the involvement of brain regions in different development stages.

The BrainSpan dataset was downloaded from the website provided in [3]. The dimension of the data matrix for our analysis is 524×43411. In this analysis, we modeled development stages and brain regions as covariates but omitted interactions between development stages and brain regions, because many combinations of development stages and brain regions only have 1 or 0 RNA-seq sample, so inference of the interaction will be unreliable. For the definition of development stages, we follow the specifications defined in the original study [3].

In the application, even though the development stages are in chronological order, we consider development stages as unordered categorical variables. The main reason is that human brain development is nonlinear, so some metagenes may only be highly active in certain developmental stages and then are leveled down or turned off. For example, the study [24]

**Table 4. The baseline RMSE and RMSE of the estimated $\hat{R}$ by INSIDER against $R$ on missing entries and samples across different simulation settings.**

| RMSE / Noise | Baseline | Missing entries | Random missing samples | Missing samples (real) |
|---|---|---|---|---|
| 0.25 | 4.5308 | 0.0460 | 0.0394 | 0.0401 |
| 0.50 | 4.6308 | 0.0907 | 0.0780 | 0.0803 |
| 0.75 | 4.5998 | 0.1397 | 0.1207 | 0.1206 |
| 1.00 | 4.4998 | 0.1827 | 0.1553 | 0.1581 |

systematically discussed the development of different brain structures across developmental stages, suggesting that the development of brain structures is specific to certain development stages. For illustration, we denote $D^{N_1 \times K}$, $T^{N_2 \times K}$, $V^{K \times M}$ as the latent representations for development stages, brain regions, and genes, respectively. There are $N_1$ development stages and $N_2$ brain regions, and $M$ is the number of genes included. The rank $K$ of latent space selected by hyperparameter tuning is 19.

## Metagenes reveal development trajectories

We adopt the convention to consider the rows of $V$ as metagenes [22]. We first looked at $D^{N_1 \times K}$, which are the levels of expression of $K$ metagenes across $N_1$ developmental stages, and computed the variance of the expression levels of each metagene across $N_1$ developmental stages (the variances are computed with the columns in $D$). We chose the most variable (16th), 2nd, and 17th metagenes. We visualized the trajectories of the selected metagenes across human brain development (Fig 2A). The expression level of the most variable metagene is high before late fetal stages and decreases to a low level after early childhood (Fig 2A). Meanwhile, we see a rapid surge in the level of the 2nd metagene between late fetal and childhood (Fig 2A). Additionally, the loading of the 17th metagene is high at the early fetal and fluctuates around zero at later stages (Fig 2A).

To decode biological mechanisms behind these metagenes, we carried out enrichment analyses to find up- (Fig 2B) and down-regulated (Fig 2C) biological processes (BPs) encoded by them. Specifically, for a given metagene $V_i$, the $i$-th row of $V$, we examine the BPs enriched by genes with large effects, that is, the genes with large absolute values for $V_i$, since these genes encode primary biological functions of the metagene. Therefore, enrichment analyses were conducted on the set of genes in the upper or lower 2.5% quantile of $V_i$ to explore up- or down-regulated BPs encoded by the metagene, respectively. Here up- or down-regulated BPs refer to BPs enriched by up- or down-regulated genes. Throughout our work, we use them to convey the meaning without loss of clarity. Other technical details for the enrichment analysis are presented in the Supplementary Content (S1 Text). The up- and down-regulated BPs for the 2nd (S1A and S1B Fig) and 17th (S1C and S1D Fig) metagenes are also provided in S1 Fig.

On the one hand, a joint analysis of the trajectories of the (16th, 17th) metagenes (Fig 2A) with their corresponding up-regulated BPs shows that the embryo is under rapid differentiation and development at the early fetal (S1C Fig) and that developments of brain regions, such as the hippocampus, pallium, and forebrain, are high before the mid-fetal period (Fig 2B). As core systems of the brain are established before birth [25], levels of BPs involved in cell development, axonogenesis, myelination, and gliogenesis fall rapidly after the late fetal period (Figs 2A and 2B and S1C). The basic structure of human brains does not undergo substantial changes after birth, so we observed a decline in levels of BPs related to the development of brain regions, such as the forebrain, telencephalon, pallium, and hippocampus (Fig 2A and 2B). With its continued development, human brains mature well into the 20s [26], corresponding to a relatively low level of these BPs in young adulthood (Fig 2A and 2B).

On the other hand, we observed that the BPs involved in learning, memory, and cognition are relatively low in fetal stages (Figs 2A and 2C and S1D). With the development of human brains, we also noticed a rapid surge in the level of BPs related to learning, memory, and cognition during infancy (Figs 2A and S1A) and a stable rise in levels of these BPs after mid-fetal stages (Fig 2A and 2C). After the maturation of human brains, these BPs maintain a high level (Fig 2A and 2C). During the development of learning, memory, and cognition, new neural networks form in the brain [27], and the level of neurotransmitters increases simultaneously [28]. Our findings also show that the BPs involved in synaptic functions and neurotransmitter

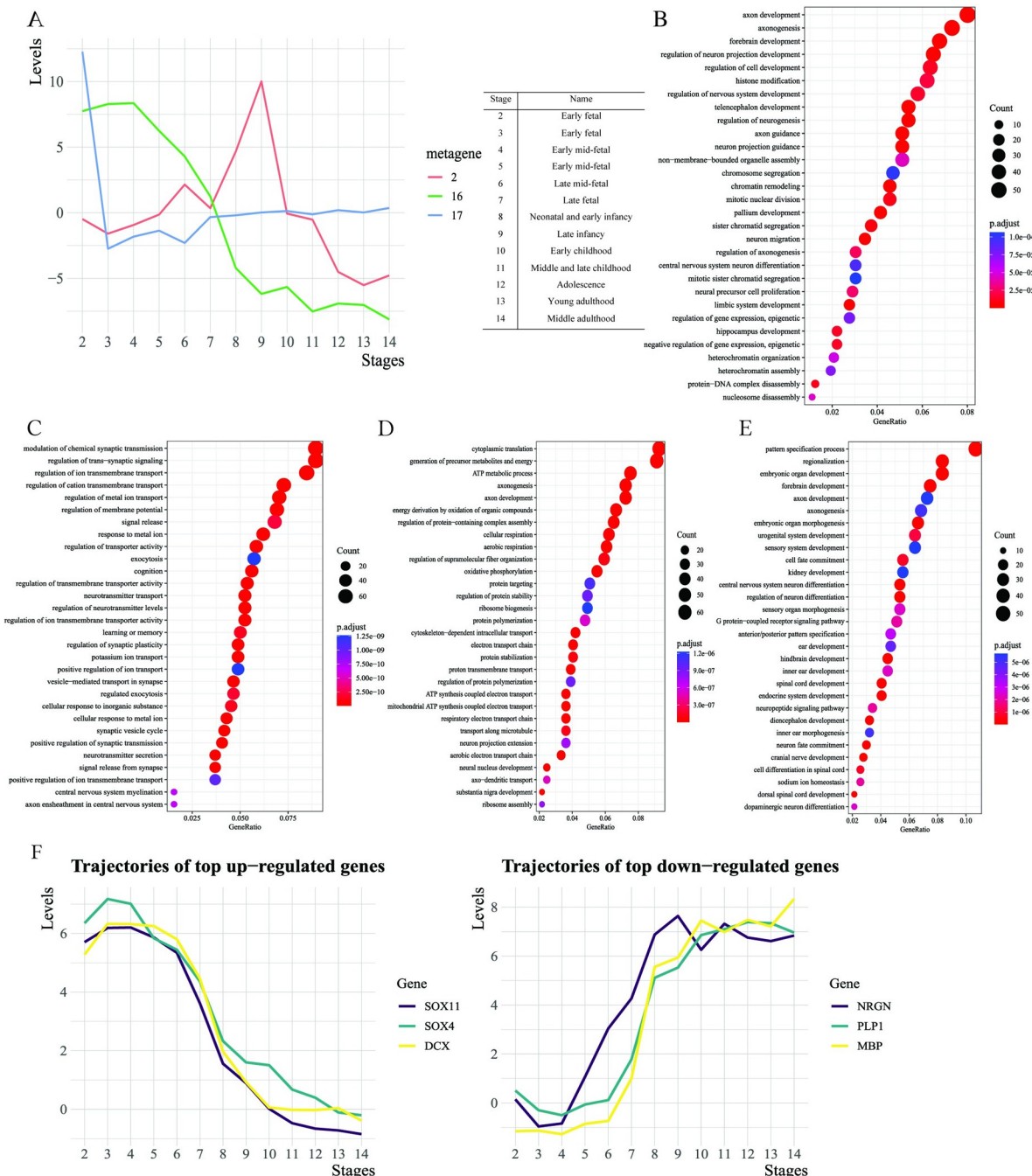

**Fig 2. Downstream analyses with results from the application to the BrainSpan data.** A. The trajectories of the selected metagene across human brain development are displayed in the left of panel A. The definition of stages is provided in the table on the right of panel A. B. The top 30 up-regulated BPs enriched by the most variable (16th) metagene is presented. C. The top 30 down-regulated BPs enriched by the most variable (16th) metagene is displayed. D. The top 30 up-regulated BPs in the primary motor and somatosensory cortex (M1C-S1C) are shown in panel D. E. The top 30 down-regulated BPs in the M1C-S1C are shown in panel E. F. Trajectories of the adjusted expression for the selected genes at the whole brain level are shown in panel F. The definition of stages in the figure is the same as in panel A.

activities follow a similar trajectory as those of learning, memory, and cognition (Figs 2A and 2C, S1A and S1D).

The trajectory of the least variable metagene is highly negative and flat across human brain development (S2A Fig). Subsequent enrichment analysis revealed that the down-regulated BPs

enriched by metagene are involved in cell and ATP metabolism and cell communication (S2B Fig), which generally maintain a high level during brain development.

## Metagenes characterize development stages and brain regions

Previously, metagenes depicted the development trajectories of human brains. Moreover, metagenes also can characterize development stages and brain regions. This can be achieved by examining BPs enriched by metagenes with highly distinct values for development stages or brain regions. We highlighted several metagenes for demonstration (Tabs A and B in S1 Table).

On the one hand, one metagene was highlighted to characterize development stages for demonstration. First, we observed that the expression level of the 5th metagene for the early mid-fetal (the 5th stage in Fig 2A) is highly positive and much greater than for other stages (Tab A in S1 Table). Subsequently, the top up-regulated BPs by the 5th metagene involve ossification and development of bone, muscle, and sensory systems (S2C Fig). The results suggest that the mid-fetal period is vital for bone and muscle formation and contributes to sensory system development.

On the other hand, we also highlighted one metagene to uncover the functions of brain regions. The loading of the 1st metagene for the striatum is highly positive. Furthermore, the absolute value of the metagene is much greater for the brain region than for other brain regions (Tab B in S1 Table). As shown by subsequent enrichment analysis, the top 30 up-regulated BPs by the 1st metagene cover learning or memory, cognition, and hormone (e.g., dopamine, monoamine, and catecholamine) secretion and transport (S2D Fig), suggesting that the striatum is responsible for human emotion and cognition (e.g., reward system) [29].

## Reveal trajectories of genes expression across human brain development at the whole-brain level

The development of the human brains is heterogenous across different individuals and not linear across different brain regions. Thus, the confounding variation from donors and tissues makes it difficult to reveal gene expression changes at the whole-brain level across human brain development. One can only show trajectories of gene expression in a tissue-wise manner, as displayed in the study [23].

However, INSIDER allows us to obtain the 'adjusted' gene expression profiles across development stages by multiplying $D$ with $V$. Note that variation from tissue and donor is controlled in the adjusted gene expression profiles, which cannot be readily computed from conventional approaches. Here we showed the trajectories of summarized expression changes for six selected genes (Fig 2F), which are chosen from genes in the top and bottom of the expression profile for the most variable (16th) metagene selected previously.

The left panel in Fig 2F showed that the trajectory of *DCX*, expressed in neuronal progenitor cells and immature migrating neurons, is similar to that of BPs involved in human brain development discussed previously and consistent with its trend in the hippocampus [23].

Regarding *Sox4* and *Sox11*, known key controllers of embryonic and fetal development, our results suggested that they are expressed at high levels in the early development stages (left panel in Fig 2F). Furthermore, they also demonstrate a similar trajectory across human brain development in the panel.

Both *MBP* and *PLP1* are involved in the myelination in the CNS, which is vital to cognition development and function and is largely completed in early adulthood [30,31]. Another gene *NRGN* regulates synaptic plasticity crucial for synaptic function [32]. The trends of trajectories

of the three genes are similar and upward (right panel in Fig 2F). This observation is consistent with the development of cognition discussed previously.

## Uncover biological mechanisms underlying the development of specific brain regions

The tissue representation from INSIDER enables us to explore BPs in specific brain regions by computing their adjusted gene expression profiles. Specifically, the adjusted gene expression profiles for brain regions were obtained by multiplying *T* with *V*. In the multiplication, two metagenes with the least variance across different brain regions were excluded because they aggregate the expression of cell metabolic and ATP metabolic processes and cell communication. Thus, excluding them enables us better uncover BPs related to brain development.

Then, we conducted enrichment analyses on the adjusted gene expression profile of M1C-S1C (primary motor and somatosensory cortex) to explore up- (Fig 2D) and down-regulated (Fig 2E) BPs for the brain region. We have no particular preference for specific brain region for further analysis; the main reason for choosing M1C-S1C for presentation is that its function is probably more specific and has less overlap with other regions, so the biological relevance of our approach can be better demonstrated.

The M1C-S1C is mainly responsible for generating signals to direct the body's movement and processing somatic sensations. The development of this region involves axonogenesis and regulation of neuron projection development (Fig 2D) and thus increases connectivity between synapses. The levels of synaptic activities also increase with the increased connectivity [33]. Therefore, it is reasonable to observe that BPs involved in synaptic activities, such as chemical synaptic transmission, trans−synaptic signaling, and synaptic plasticity, are upregulated in this brain region (Fig 2D). However, the levels of gene expression for BPs involved in the development of human organs, such as the kidney, renal system, spinal cord, and sensory organ (e.g., inner ear and ear), are low in the brain region (Fig 2E).

In summary, through the analysis of the BrainSpan data, the advantages of INSIDER have been demonstrated. It enables analyzing gene expression data that cannot be readily formatted into 3D tensors, as there could be several observations for some combinations of brain regions and developmental stages. Our approach reveals the brain developmental trajectories by metagenes, and up- and down-regulated BPs encoded by the metagenes. Our method also characterizes the molecular basis underlying neurodevelopment for different brain regions and developmental stages by examining their underlying up- and down-regulated BPs. However, standard PCA- and especially GLM-based methods are unapplicable for these tasks in RNA-seq data analysis. Moreover, NMF may not be readily applicable as it is not straightforward to directly infer down-regulations of gene expression.

## Analysis of the aging, dementia, and TBI data

Dementia is a kind of neurodegenerative disorder, and there are various underlying causes. Traumatic brain injury (TBI) may lead to dementia, but the most common cause is Alzheimer's disease (AD). The Aging, Dementia, and TBI Study aims to characterize neuropathologic, molecular, and transcriptomic changes in the brains of controls and TBI cases from an aged population-based cohort [34], which offers RNA-Seq data of temporal cortex (TC), parietal cortex (PC), cortical white matter (WM), and hippocampus (HPC) of controls and TBI cases.

We downloaded the aging, dementia, and TBI data from the website provided in [34]. The dimension of our dataset is $377 \times 44,477$, and the RNA-seq samples come from 107 donors. In this analysis, we seek to explore BPs contributing to the progression of dementia and discover BPs contributing to dementia in specific brain structures. Here we modeled four covariates:

donor, brain structure, diagnosis of dementia, and gene, and incorporated interactions between brain structures and phenotype (dementia diagnosis). The number of ranks $K$ of latent space selected by hyperparameter tuning is 23.

## Clustering analysis of donor representations

It is challenging to perform clustering analysis in high-dimensional settings, for example, with high-dimensional omics data. However, INSIDER enables us to cluster donors with donor representations to characterize donors and even identify disease subtypes, which is in line with personalized treatment delivery. In practice, we performed hierarchical clustering with latent donor representations (Fig 3A) without the least variable metagene, which encodes BPs related to ATP metabolic processes. We selected two as the number of clusters. Subsequently, we examined the clinical or demographic relevance of the clusters. The two-tailed standard ($P = 0.0251$) and rank-based ANOVA tests ($P = 0.0185$) indicate that the clustering memberships statistically correlate with donor age distribution. We note that age itself was not included as an input for clustering, yet our clustering primarily based on expression data is able to identify subgroups with important clinical differences (e.g., age).

We also explored whether donor representations enable us to reveal subgroups related to subtypes of dementia. Since not all metagenes are associated with dementia, we filtered off several metagenes that have little relevance to dementia by examining the relevance of the top 30 up- and down-regulated BPs to dementia by searching literatures. Then, we performed hierarchical clustering on donor representation with these kept metagenes. The dendrogram (S3A Fig) indicates three subgroups.

Two-tailed standard ($P = 0.0205$) and rank-based ($P = 0.0731$) ANOVA tests yielded similar $p$-values when we examined the association between the clustering and age distribution. In further exploring its relevance to the clinical diagnoses of dementia, Fisher's exact tests revealed that clusters have a statistically significant association with the Braak staging of dementia ($P = 0.0025$) and showed a trend of association with diagnoses of dementia by the NINCDS-ADRDA Alzheimer's criteria ($P = 0.108$), one of the most widely used criteria in the diagnosis of Alzheimer's disease. Note that clinical or pathological features related to staging are not used as input for the clustering, hence this serves as an 'external validation' of the validity of the clusters. In other words, our proposed clustering framework, primarily based on expression profiles with dimension reduction by INSIDER, might indeed be useful in identifying clinically distinct and meaningful subgroups, which are consistent with widely used clinical staging/diagnosis systems.

## Metagenes reveal the biological mechanism of dementia

Here we explored the BPs significantly down-regulated in dementia compared with controls to shed light on the mechanism of dementia. In practice, we selected the top 2 variable metagenes in phenotype representation, as they achieved the most significant $p$-values in the following enrichment analysis. Then, the adjusted gene expression profiles for dementia and control were obtained by multiplying phenotype representations by gene representation with only the selected metagenes. Subsequently, we examined down-regulated BPs enriched by the difference in gene expression between dementia and control (Fig 3B).

The top 30 BPs enriched for down-regulated genes are statistically significant at the $p < 5e$-$5$ level. We observed that expression levels of genes for BPs involved in cognition, learning, and memory were significantly lower in the dementia brain than in the control brain (Fig 3B). In addition, significant differences in BPs related to synaptic activities, such as trans-synaptic signaling, and chemical synaptic transmission, were also observed when comparing dementia

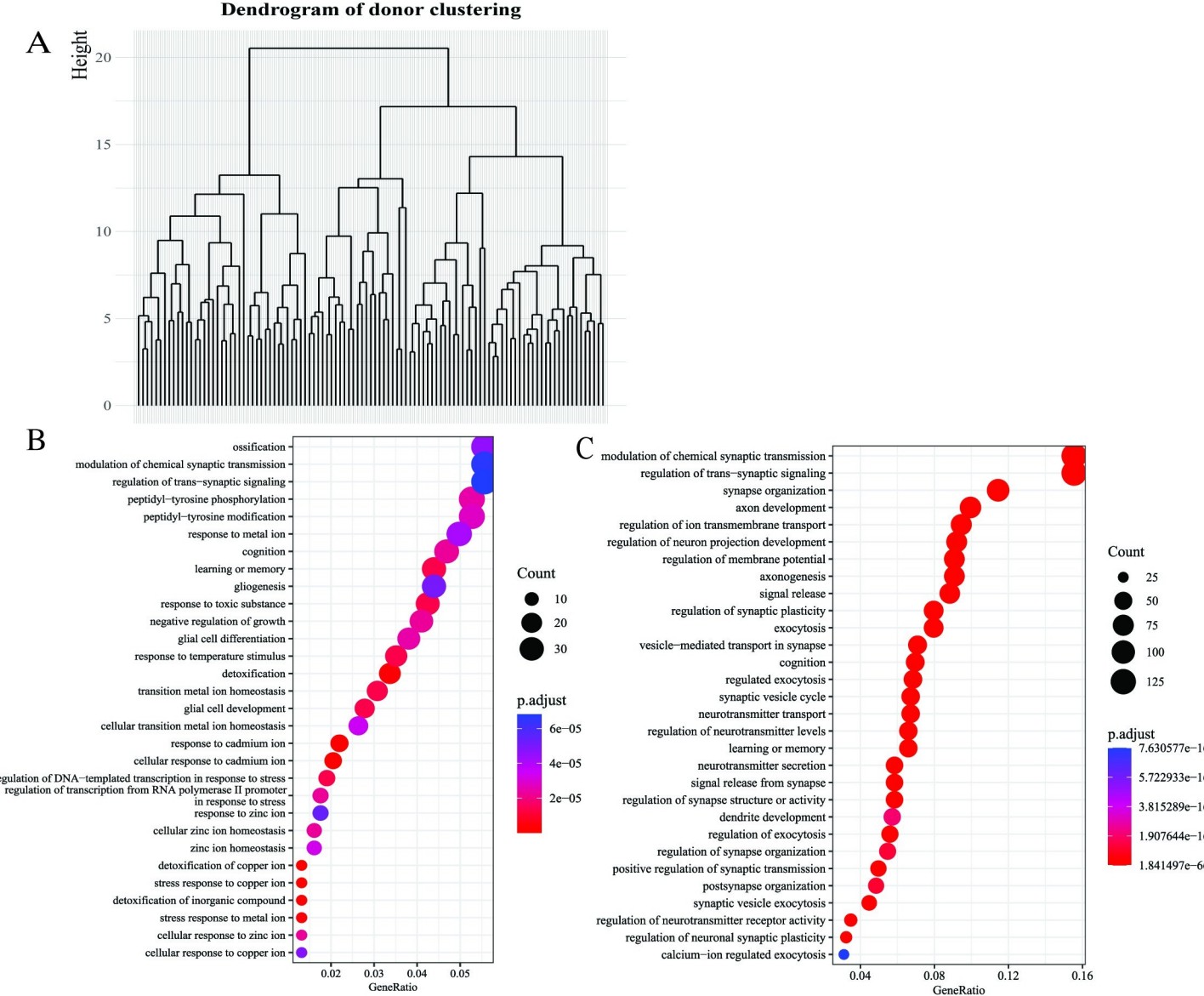

**Fig 3. Downstream analyses with results from the application to the aging, dementia, and TBI data.** A. The figure shows the dendrogram of hierarchical clustering with donor representations. B. The top 30 down-regulated BPs enriched for the difference in gene expression between dementia and control are displayed. C. The top 30 down-regulated BPs enriched by the difference in gene expression of left HPC between dementia and control are presented.

and controls (Fig 3B). This observation is supported by previous studies [35–37], suggesting that synaptic signaling and transmission play indispensable roles in the nervous system function, and their dysfunctions might be important contributing factors to dementia.

Interestingly, expression levels of genes involved in ossification (bone tissue formation) are also significantly different between dementia and control (Fig 3B). The connection between impaired bone microarchitecture or regeneration and dementia has been reported [38]. Moreover, a degree of comorbidity of AD and osteoporosis is also supported by epidemiological studies [39]. On the other hand, several BPs related to metal ions are also identified in the enrichment analysis (Fig 3B), suggesting that expression levels of genes involved in these BPs are lower in dementia than in normal. Since metal ions play essential roles in neurotransmitter

synthesis and energy metabolism, their deficiencies may contribute to cognitive decline and memory loss [40], which are the most common symptoms of dementia.

## Metagenes characterize donors and functions of brain regions

In addition to revealing the mechanism of dementia, metagenes can help us to characterize donors and reveal functions of brain regions among the aged donors.

We first employed standard two-tailed t-tests to identify metagenes associated with donor characteristics. Statistical test results showed the most statistically significant difference in loadings for the 13th metagene between donors who experienced TBI with loss of consciousness (LOC) and those not experiencing TBI ($P = 0.0242$). Subsequent enrichment analysis with the 13th metagene reveals that the top 30 down-regulated BPs involve cognitive functions (S3B Fig), suggesting that experiencing TBI with LOC may be associated with cognitive dysfunction.

For demonstration, we highlighted one metagene (17th) to reveal the functions of the white matter (WM) (S2 Table) of the forebrain. The loadings of the 17th metagene are negative only for WMs of both the left and right forebrain (S2 Table). Subsequent enrichment analysis with the 17th metagene revealed that the top 30 up-regulated BPs involve learning, memory, cognition, and synaptic functions (S3C Fig).

This suggests that expression of genes involved in these BPs is relatively low in the WM of both the left and right forebrains, compared to other brain regions. Moreover, we noticed that the loading of this metagene is lower in the WM of the left forebrain than in its counterpart. However, this cannot help us lead to the claim that the WM of the left forebrain is less involved in these BPs than its counterpart, as a single metagene may not show the whole picture of the involvement of WMs.

To further clarify the issue, we examined the top down-regulated BPs in WMs of the left and right forebrain (S3E and S3F Fig, respectively). The BPs down-regulated in WM of the left forebrain (S3E Fig) are slightly more statistically significant than those in the right counterpart (S3F Fig). On the other hand, the BPs down-regulated in both areas are basically the same. Our analysis suggests that both areas of the WM may similarly involve in dementia.

## Incorporating the interaction improves INSIDER

To verify that the performance of INSIDER is significantly improved by considering interaction, we conducted the following statistical test. Generally, performance on held-out data is considered a golden standard to evaluate model performance. Thus, we examine whether incorporating interaction terms improves the performance of INSIDER on held-out data using RMSE. First, we randomly drew 10% of the elements without replacement from the data matrix as testset and considered the left elements as trainset. Second, we followed the same procedure to select hyperparameters for INSIDER with and without the interaction term. Then, we ran INSIDER on the trainset with the selected hyperparameters 50 times for 30 iterations with different parameter initializations and reported their RMSE on the testset. Finally, we examined whether the RMSE of INSIDER with interaction is statistically significantly lower than that of INSIDER without interaction with a one-sided $t$-test. The $p$-value ($p$-value $< 2.2e\text{-}16$) from the test suggests that incorporating interaction improves the performance of INSIDER. The mean with standard error for INSIDER with and without interaction are $0.2297+0.0001$ and $0.2302+0.0001$, respectively. To our knowledge, the relatively small improvement in RMSE is due to the fact that only a small proportion of genes contribute to the interaction between dementia and brain structures.

### Discover heterogeneous effects of dementia on brain structures

Here we further show that the interaction term improves model interpretation of INSIDER. As dementia was shown to have potentially heterogeneous effects on the left and right HPC at gene levels (Section II in S1 Text and S4 Fig), we seek to compare BPs down-regulated by dementia in the left and right HPC to clarify the issue further.

Here we will highlight how INSIDER with interaction is useful in unraveling interactions between biological variables (in this case, dementia diagnosis × brain structure). Compared to controls, dementia is associated with changes in gene expression, but the degree of change may not be uniform across different brain structures/regions. For instance, dementia may increase the expression of gene $X$ by 1 unit, and the expression is also, on average, elevated by 1 unit across brain regions. However, dementia could be associated with a 5-unit increase in gene $X$ expression in region $Y$, due to the presence of an interaction.

First, we used the left HPC as an example to show how to obtain BPs down-regulated by dementia. First, we multiplied the submatrix $W_k$ of interaction representation corresponding to the left HPC by gene representation $V$. In the operation, several metagenes with the least variance in the submatrix were excluded. Technical details of metagene selection are provided in S1 Text. Then, we examined the down-regulated BPs enriched by genes showing significant differences in expression in the left HPC between dementia and controls (Fig 3C). Similarly, BPs down-regulated by dementia in the right HPC can be obtained (S3D Fig).

Overall, most down-regulated BPs are more significant in the left HPC ($P$ at 1e-66 in Fig 3C) than in the right ($P$ at 2e-57 in S3D Fig). Moreover, the gene ratio in Fig 3C is also greater than in S3D Fig. Here gene ratio is the proportion of genes enriched for a BP. Regarding BPs enriched for the two brain structures, there is little difference in general. This leads to a similar finding that the left and right HPC may be affected differently by dementia and supported by previous studies we discussed earlier.

To summarize, INSIDER demonstrates the following advantages over existing methods in analyzing the aging, dementia, and TBI data. INSIDER enables clustering donors into disease subtypes. This task is inapplicable to GLM-based methods for DGE analysis. Moreover, INSIDER reduces dimension with respect to one variable (e.g., donor) while controlling for variation from other variables (e.g., brain regions) and helps gain insights into the interaction between brain regions and dementia diagnosis. However, PCA- and NMF-based methods are also not applicable for these tasks, as they cannot perform dimension reduction with respect to different covariates separately and consider the interaction between covariates. Moreover, tensor decomposition cannot be employed to explore the interaction, as there are several observations for different combinations of brain regions and diagnosis of dementia and thus the data cannot be formatted into order-3 tensors.

### Computing time and memory requirements

The computing time of INSIDER varies depending on the stopping criteria and computational resources allocated. In our study, with the defined stopping criteria, INSIDER takes approximately 5 hours and 8 hours to run the BrainSpan dataset (524×43,411) and the aging, dementia, and TBI dataset (377×44,477), respectively, when using 30 cores (Intel Xeon Gold 6230 CPU @2.10GHz).

Owing to our implementation with RcppArmadillo for parallel computing, INSIDER has relatively low memory requirements. It requires less than 5GB of memory for the BrainSpan dataset and the aging, dementia, and TBI dataset.

## Discussion

We demonstrated the versatility and potential of INSIDER through applications in which various downstream analyses were performed. In the first application, the development trajectories across the human life span were revealed, and up- and down-regulated BPs underlying them were uncovered. Moreover, the implicated brain regions were characterized. In the other application, we explored the mechanisms contributing to dementia, identified subgroups among donors with latent donor representations, and examined the contribution of tissues (e.g., brain regions) to biological conditions (e.g., dementia). Our results are supported by the literature. For example, the interpretation of the development trajectories is consistent with findings from previous studies.

INSIDER has the following advantages. First, it is general and flexible, in which covariates and interactions in modeling can be adjusted according to research interest. Second, it is versatile and interpretable in terms of downstream analysis. Various downstream analyses were carried out, such as revealing development trajectory, clustering donors, revealing disease mechanisms, and exploring interactions between covariates. Third, it encourages sparsity in gene representations by introducing elastic-net regularization, thus facilitating result interpretations. Fourth, it can capture interactions between covariates, common in biological settings. Finally, it has good scalability and allows missing values.

Moreover, the advantages of INSIDER over existing methods have been demonstrated via applications. Compared with INSIDER, GLM-based methods for DGE analysis do not aim for dimension reduction and thus do not enable clustering donors in low-dimension, revealing developmental trajectories, and characterizing brain regions or development stages. Moreover, traditional PCA/factor analysis decomposes the variation of $Z^{N \times M}$ by $Z^{N \times M} = H^{N \times K} L^{K \times M}$. Both $H^{N \times K}$ and $L^{K \times M}$ are unobserved. The $L$ combines two or more variables into a single component/factor. Similarly, INSIDER combines several genes that tend to be co-expressed or work together to induce biological functions into a single metagene. Thus, $V$ in INSIDER is analogous to $L$ in factor analysis, the elements of $V$ represent the effects of metagenes on individual genes, and the contribution of individual genes to each metagene is represented by the corresponding row of $V$. Moreover, INSIDER can be considered a generalization/extension of factor analysis, where the factors are modelled as being under the influence of some covariates. In INSIDER, we introduce a model of $F$ in terms of covariates $X$ by $F = XB$, where $X$ is an indicator matrix of measured covariates and $B$ is a matrix of coefficients. The covariates are classified into donor ($X_D$), phenotype ($X_P$), tissue ($X_T$), and phenotype-by-tissue interactions ($X_W$). However, these indicator matrices can be joined into a single matrix $X = (X_D, X_P, X_T, X_W)$, if the coefficient matrices are also joined together into a single matrix $B = (D^\mathsf{T}, P^\mathsf{T}, T^\mathsf{T}, W^\mathsf{T})^\mathsf{T}$. Thus, this gives the overall INSIDER model by $Z = XBV + \Theta$. Different from PCA/factor analysis, INSIDER directly associates the covariates ($X_D, X_P, X_T, X_W$) with the metagenes through the matrices $D, P, T, W$, and this provides insights on how covariates affect the biological pathways. However, the estimation of $V$ in PCA/factor analysis is not guided by the covariates and thus it cannot directly associate the components/factors with the covariates. In addition, INSIDER enables computing adjusted expression profiles for covariates, thus uncovering the effect of covariates on gene expression. In contrast with INSIDER, standard NMF-based methods cannot model variation from *multiple* variables separately and capture their interaction. and unable directly infer down-regulation in biological settings, which impairs its interpretability. Compared with tensor decomposition, the closest analog of INSIDER, INSIDER also demonstrates its superiority through simulation studies and is able to handle the situation where the data matrix cannot be formatted into tensors, as demonstrated via applications. A more

explicitly comparison of the above capabilities of INSIDER with those of the other methods is displayed in S3 Table.

Bulk RNA-Seq techniques are popular in studying tissue-specific expression profiles. Usually, confounding variables (covariates) are inevitably introduced in the RNA-Seq expression data. For example, GTEx [2] data contains confounding variables and other features leading to variations in expression, such as donor, gender, and age. However, few studies fully addressed the issue of *multiple* confounders or covariates while also achieving dimensionality reduction (DR). Existing approaches such as PCA, factor analysis, and non-negative matrix factorization only can reduce data dimensionality but have substantial limitations, as described above. The importance of an improved methodology to perform DR in high-dimensional data with *multiple* covariates has become more prominent with the increasing availability of RNA-seq data and the popularity of joint data analysis from different sources. For example, the PsychEN-CODE project [4] is composed of several independent projects, such as ATP (the Autism Tissue Program) [5], CMC (the CommonMind Consortium) [41], and BrainGVEX [4], with different study aims. INSIDER well addresses the challenge and takes account of sparsity in gene expression and the interaction between covariates simultaneously. More importantly, it has excellent interpretability and can be applied to large-scale data analysis.

Meanwhile, we notice that the development stages are in chronological order. In practice, we consider them as unordered categorical variables because human brain development is nonlinear and the development of brain structures is specific to certain development stages, as we discussed previously. It is more reasonable to model development stages as unordered categorical variables. On the other hand, our simulation studies show that INSIDER can recover the monotonic trend for ordinal variables (Section VI in S1 Text). Therefore, it is a feasible solution for INSIDER to consider ordinal variables as unordered categorical variables. It is ideal for INSIDER to consider ordinal variables explicitly. We would like to address this issue in future.

Future works may also include further analysis of various bulk RNA-seq datasets and extend it to single-cell data analysis. In addition, further experimental studies to validate the findings from our applications will be an important future direction. We note that the biological findings should be considered tentative until further replicated by further experimental and clinical studies.

## Conclusion

This article proposed a novel statistical approach, INSIDER, based on matrix decomposition, to model the variation contributed by confounding variables or covariates in bulk RNA-Seq data and decompose it into a shared low-rank latent space. The variation may originate differently, such as donor, gender, tissue, or other biological conditions. INSIDER is a general and flexible framework, which is robust to missing values. In particular, it can capture interactions between variables or covariates. Moreover, it encourages sparsity in latent representations by imposing the elastic net penalty, thus facilitating model interpretations. In our applications, various downstream analyses were carried out based on results from INSIDER. The broad applicability of INSIDER in biomedical and clinical settings was demonstrated in the applications.

## Methods

Here we propose INSIDER, based on additive matrix factorization, to decompose heterogeneous biological variation in RNA-Seq data into a low-rank latent space.

## Model specifications

Denote $z_{ijhm}$ as the expression level of gene $m$ of sample $s$ from tissue $h$ of donor $i$ with phenotype $j$. We assume that there are $N$ observations, $N_1$ donors, $N_2$ phenotypes, and $N_3$ tissues. $z_{ijhm}$ is modelled as

$$\hat{z}_{ijhm} = d_i^T v_m + p_j^T v_m + t_h^T v_m + w_{jh}^T v_m + u_s^T v_m \tag{1}$$

where $d_i, p_j, t_h, w_{jh}, u_s, v_m$ are vectors of length $K$, and $u_s$ is usually ignored when donor, tissue, and phenotype information is available and can be included to capture variation from unknown sources when part or all of the source information is unavailable. Note that the above equation incorporates the interaction effect between tissues and phenotypes via introducing $w_{jh}$ on gene expression. For example, a certain disease may be associated with changes in gene expression, but the degree of change may differ by the tissue type. The objective function that we seek to minimize for Eq 1 is defined as

$$\mathcal{L}(d,p,t,w,u,v) = \frac{1}{2}\sum_{i,j,h,m}\left[z_{ijhm} - \left(d_i + p_j + t_h + w_{jh} + u_s\right)^T v_m\right]^2$$

$$+ \frac{1}{2}\lambda\left[\sum_i||d_i||_2^2 + \sum_j||p_j||_2^2 + \sum_h||t_h||_2^2 + \sum_{j,h}||w_{jh}||_2^2 + \sum_s||u_s||_2^2\right] \tag{2}$$

$$+ \lambda\left[\frac{1}{2}(1-\alpha)\sum_m||v_m||_2^2 + \alpha\sum_m|v_m|_1\right]$$

The Eq 2 can be represented with matrix notations. Let $Z \in \mathbb{R}^{N \times M}$ represent the data matrix for the expression profile of $N$ samples and $M$ genes. The matrices $D^{N_1 \times K}, T^{N_2 \times K}, P^{N_3 \times K}, W^{(N_2 * N_3) \times K}, U^{N \times K}, V^{K \times M}$ are the rank $K$ latent representations for $N_1$ donors, $N_2$ phenotypes, $N_3$ tissues, $N_2 * N_3$ interactions, $N$ observations, and $M$ genes, respectively. $X_D^{N \times N_1}, X_T^{N \times N_2}, X_P^{N \times N_3}, X_W^{N \times (N_2 * N_3)}, X_U^{N \times N}$ are indicator matrices, representing the dummy design matrices for $N$ samples. $X_U^{N \times N}$ is an identity matrix, which can be ignored and is added for the consideration of consistency in notations. Eq 2 can be rewritten as:

$$\mathcal{L}(D,P,T,W,U,V) = \frac{1}{2}\|Z - (X_D D + X_P P + X_T T + X_W W + X_U U)V\|_F^2 +$$

$$\frac{1}{2}\lambda\left[\| D \|_F^2 + \| P \|_F^2 + \| T \|_F^2 + \| W \|_F^2 + \| U \|_F^2\right] +$$

$$\lambda\left[\frac{1}{2}(1-\alpha) \| V \|_F^2 + \alpha|V|_1\right].$$

Missing entries in $Z$ are removed in the summation in the first line of the equation.

## Model fitting

Alternating block coordinate descent (BCD) was employed to optimize Eq 2. In each iteration, we update all parameters of INSIDER sequentially. We repeat the process until the stop criteria meets.

## Optimize the objective function

First, we choose to maximize the objective function with respect to $p_j$ in Eq 2. By fixing all $d_i, t_h$, $w_{jh}, u_s, v_m$ and taking the derivative with respect to $p_j$, we can have the update for $p_j$ as follows

$$p_j = \left[ N_j V V^\top + \lambda \mathbb{I}_K \right]^{-1} \sum_{k \in S_j} V \tilde{z}_k,$$

where $\tilde{Z} = Z - (X_D D + X_T T + X_W W + X_U U) V$, $S_j$ is the set of row indices of samples with phenotype $j$, and $N_j$ is the number of elements in $S_j$. Similarly, the update for $t_h$ can be derived as

$$t_h = [N_h V V^\top + \lambda \mathbb{I}_K]^{-1} \sum_{k \in S_h} V \tilde{z}_k,$$

where $\tilde{Z} = Z - (X_D D + X_P P + X_W W + X_U U) V$, $S_h$ is the set of row indices of samples from tissue $h$, and $N_h$ is the number of elements in $S_h$. Likewise, we can have the update for $d_i$:

$$d_i = [N_i V V^\top + \lambda \mathbb{I}_K]^{-1} \sum_{k \in S_i} V \tilde{z}_k,$$

where $\tilde{Z} = Z - (X_T T + X_P P + X_W W + X_U U) V$, $S_i$ is the set of row indices of samples from donor $i$, and $N_i$ denote the number of elements in $S_i$. Besides, the update for $w_{jh}$ can be derived as follows

$$w_{jh} = \left[ N_{jh} V V^\top + \lambda \mathbb{I}_K \right]^{-1} \sum_{k \in S_{jh}} V \tilde{z}_k,$$

where $\tilde{Z} = Z - (X_D D + X_T T + X_P P + X_U U) V$,, $S_{jh}$ is the set of row indices of samples of tissue $h$ with phenotype $j$, and $N_{jh}$ is the number of elements in $S_{jh}$. Finally, the update for $u_s$ can be derived as follows

$$u_s = [V V^\top + \lambda \mathbb{I}_K]^{-1} V \tilde{z}_s,$$

where $\tilde{Z} = Z - (X_D D + X_T T + X_P P + X_W W) V$, $\tilde{z}_s$ is the $s$-th row of $\tilde{z}_s$, and $u_s$ is latent representation for the $s$-th sample.

In the above, we derived the updates for $D, P, T, W, U$ for categorical covariates. In fact, INSIDER can also handle continuous covariate. For illustration, let denote $q$ a vector of length $K$ for latent representation for a continuous covariate, and $x_q^{N \times 1}$ is the variable across $N$ observations.

$$\mathcal{L}(D, q, T, W, U, V) = \frac{1}{2} \| Z - \left( X_D D + x_q q^\top + X_T T + X_W W + X_U U \right) V \|_F^2$$

$$+ \frac{1}{2} \lambda \left[ \| D \|_F^2 + \| q \|_2^2 + \| T \|_F^2 + \| W \|_F^2 + \| U \|_F^2 \right]$$

$$+ \lambda \left[ \frac{1}{2} (1 - \alpha) \| V \|_F^2 + \alpha |V|_1 \right].$$

In the equation, the meaning of other notations is the same as in the manuscript. With

other parameters fixed, the objective function with respect to $q$ can be simplified as follows:

$$\mathcal{L}(q) = \text{argmin}_q \frac{1}{2} \text{Tr}\left[ \left( \tilde{Z} - x_q q^\top V \right)^\top \left( \tilde{Z} - x_q q^\top V \right) \right] + \frac{1}{2} \lambda \parallel q \parallel_2^2,$$

where $\tilde{Z} = Z - (X_D D + X_T T + X_W W + X_U U) V$. By minimizing the objective with respect to $q$, the update for $q$ can be obtained by

$$q = \left( x_q^\top x_q V V^\top + \lambda \mathbb{I}_k \right)^{-1} V \tilde{Z}^\top x_q.$$

This feature has been incorporated into the software of INSIDER.

When we seek to minimize the objective function defined by Eq 2 with respect to $v_m$, the $m$-th column of $V$, with all the other parameters fixed, the equation can be simplified into the following elastic net problem

$$\mathcal{L}(v_m) = \frac{1}{2} \| Z_m - E v_m \|_2^2 + \frac{1}{2} \lambda (1 - \alpha) \| v_m \|_2^2 + \lambda \alpha |v_m|_1 \qquad (3)$$

Here $E = X_D D + X_P P + X_T T + X_W W + X_U U, Z_m, Z_m$ is the expression levels of the $m$-th gene across the $N$ samples, and missing entries in $Z_m$ are removed in the calculation. The randomized coordinate descent (RCD) algorithm proposed in Algorithm 1 in S1 Text is employed to optimize each column of $V$ in parallel.

In practice, optimizing the subproblems defined by Eq 3 is computationally intensive. The safe rules for screening are adopted to accelerate the computation, which discard variables with coefficients that are shrunk to zero in optimization. To illustrate with Eq 3, the rules defined by Eq. 24 in the study [42] for discarding predictor $j$ is

$$|u_j^T Z_m| < \alpha (2\lambda - \lambda_{\max}), \qquad (4)$$

where $\lambda_{\max} = \max_j |u_j^T Z_m|$, and $u_j$ is the $j$-th column of $U$.

## Initialization, hyperparameter tuning and the stopping criterion

Regarding initialization of $D$, $P$, $T$, $W$, $U$, $V$ defined in the Eq 2, they are initiated from normal distribution $N(0, 0.001)$. We consider the solution from the previous iteration as warm start for the subproblem defined by Eq 3 in optimization.

In model selection, we use the following procedure to select the hyperparameters $\{K, \lambda, \alpha\}$ for INSIDER. Specifically, we first randomly draw 10% of elements without replacement from the data matrix as testset and consider the left data as trainset. Only the trainset is used to train INSIDER, and the testset is independent from model training and utilized for model selection. Specifically, we first train INSIDER on the trainset and then examine whether our approximations for the testset by the trained model can well approximate their real values. This practice is also consistent with the idea of cross-validation, which performs model selection based on model performance on held-out data. In fact, it is also widely used in recommendation systems [43] and matrix completion [44]. To evaluate the performance of INSIDER with different hyperparameters, the root mean squared error (RMSE) on the testset is calculated, which measures the root averaged squared difference between the approximations from INSIDER and the real values, so a lower RMSE on the testset corresponds to a better model.

In our applications, $K$ was chosen from a sequence from 10 to 30 with step size 2. The range of $K$ is enough in general and can be broadened to consider more complex models. For each try of $K$ from the sequence, we run INSIDER for 30 iterations. In the process, $\lambda$ is set to 0.1 to avoid singularity in matrix inverse, and $\alpha$ is fixed to 0. Then, we select the rank $K$

corresponding to the lowest RMSE on the testset. Empirical evidence shows that 30 iterations are sufficient to give consistent selection results, compared with 50 iterations (Section IV in S1 Text). Also, previous studies [14,22] utilized a small number of iterations (e.g., 20 iterations) to select the rank *K* for the methods based on matrix factorization. Note that our method for selecting *K* is quite computationally efficient, since we set $\alpha = 0$ and the update for all parameters can be computed in a closed form.

After choosing *K*, we define a parameter grid for $\{\lambda,\alpha\}$. For each set of candidate hyperparameters $\{\lambda,\alpha\}$, we also run alternating BCD for 30 iterations and choose the parameters with the best performance on the test set.

In our study, the stopping criteria is defined as

$$\frac{|\mathcal{L}_i(\cdot) - \mathcal{L}_{i-1}(\cdot)|}{\mathcal{L}_{i-1}(\cdot)} < \delta, \tag{5}$$

where $\mathcal{L}_i(\cdot)$ denotes the loss at the *i*-th iteration, and $\delta$ is a predefined threshold and set to $10^{-10}$ in our experiment.

## Supporting information

**S1 Text. Supplementary Content.** It provides technical details on enrichment analysis, gene selection in comparing expression differences between dementia and control across brain regions, and metagene selection for discovering heterogeneous effects of dementia on brain structures. Regarding the model selection of INSIDER, it demonstrates the effect of different numbers of iterations (e.g., 30, 50) on model selection. It also shows the capability of INSIDER to handle ordinal variables via simulation studies. Regarding the difference between INSIDER and PCA/factor analysis or GLM-based approaches, it presents an in-depth discussion on their differences. To demonstrate its wide applicability, results from analyzing gender differences in gene expression of the human brain with GTEx using INSIDER are provided. Further, derivations for random coordinate descent for elastic net problems and the RCD algorithm (Algorithm 1) for the elastic net with screening rules are also presented.
(DOCX)

**S1 Fig. The top 30 up- and down-regulated BPs enriched by the 2nd and 17th metages.** The top 30 up-regulated BPs (S1A Fig) enriched by the 2nd metagene and down-regulated BPs (S1D Fig) by the 17th metagene are related to learning, memory, cognition, and synaptic functions and neurotransmitter activities. The top 30 down-regulated BPs (S2B Fig) enriched by the 2nd metagene involve axon development, axonogenesis, myelination, and gliogenesis. The top 30 up-regulated BPs (S1C Fig) by the 17th metagene cover the BPs related to cell development and differentiation.
(TIF)

**S2 Fig. The trajectory of the least variable metagene and top 30 BPs enriched by metagenes.** S2A Fig shows that the trajectory of the least variable metagene is negative and basically flat across human brain development. The top 30 down-regulated BPs (S2B Fig) encoded by the metagene involve cell and ATP metabolism and cell communication. Moreover, the top 30 up-regulated BPs (S2C Fig) enriched by the 5th metagene involve ossification and development of bone, muscle, and sensory systems, and the top 30 up-regulated BPs (S2D Fig) by the 1st metagene cover learning or memory, cognition, and hormone (e.g., dopamine, monoamine, and catecholamine) secretion and transport.
(TIF)

**S3 Fig. Supplement Figs for analysis of the aging, dementia, and TBI data.** S3A Fig shows the dendrogram of hierarchical clustering on donor representation with the selected meta-genes. We selected 3 subgroups for further analysis. The top 30 down-regulated BPs (S3B Fig) enriched by the 13th metagene involve cognitive functions, and the top 30 up-regulated BPs (S3C Fig) enriched by the 17th cover learning, memory, cognition, and synaptic functions. S3D Fig shows the top 30 down-regulated BPs enriched by the difference in gene expression of right HPC between dementia and control. Overall, the *p*-values for most down-regulated BPs in S3D Fig are significant at 2e-57, which are greater than those for the left HPC shown in Fig 3C from the manuscript (*P* at 1e-66). Moreover, the gene ratio in S3D Fig is also smaller than in Fig 3C. This leads to the finding that the left and right HPC may be affected differently by dementia. S3E and S4F Figs show the top 30 down-regulated BPs enriched by the difference in expression profiles for WM of the left and right forebrain, respectively. The BPs in S3E Fig are slightly more statistically significant than those in S3F Fig and the BPs in the two figures are basically the same.
(TIF)

**S4 Fig. Comparison of differences in the expression of three selected genes between dementia and control across brain regions.** The figure compares the expression levels of three selected genes (NRGN, CAMK2A, and SHISA6) between dementia and control across brain regions, suggesting that dementia has potentially heterogeneous effects on the left and right HPC at gene levels.
(TIF)

**S1 Table. Tab A. The loadings of all metagenes across brain development stages in the analysis of the BrainSpan dataset.** The table shows that the loading of the 5th metagene for the early-mid fetal stage (the 5th stage) is highly positive, and its absolute value is much larger than that of the metagene for other developmental stages. Thus, the 5th metagene can be used to characterize the stage. **Tab B. The loadings of all metagenes across all brain regions in the analysis of the BrainSpan dataset.** The table shows that the loading of the 1st metagene for the striatum is highly positive, and its absolute value is much larger than that of the metagene for other brain regions. Thus, the 1st metagene can be used to characterize the brain region.
(XLSX)

**S2 Table. The loadings of all metagenes across all brain regions in the analysis of the aging, dementia, and TBI dataset.** The table shows that the loadings of the 17th metagene are only negative for FWM (both left and right) and their absolute value is much larger than those for other brain regions. Therefore, this metagene can be utilized to reveal the functions of FWM in the brains of aged donors.
(XLSX)

**S3 Table. Comparison of INSIDER with the other methods in their capabilities in accommodating different situations in the analysis of gene expression data.** In the table, we compare INSIDER with other methods, including PCA/NMF, SDA, and the GLM-based methods for DEG analysis, in their abilities to handle more than 3-dimensional data, to incorporate interaction, to allow missing data, to reduce data dimensionality, to perform differential expression analysis, and to handle multidimensional ($\geq 3$ dimensions) data that cannot fit into tensors.
(XLSX)

## Author Contributions

**Conceptualization:** Kai Zhao, Pak Chung Sham.

**Data curation:** Kai Zhao.

**Formal analysis:** Kai Zhao.

**Funding acquisition:** Zhixiang Lin.

**Investigation:** Pak Chung Sham, Hon-Cheong So, Zhixiang Lin.

**Methodology:** Kai Zhao, Pak Chung Sham, Zhixiang Lin.

**Project administration:** Zhixiang Lin.

**Software:** Kai Zhao, Sen Huang.

**Supervision:** Hon-Cheong So, Zhixiang Lin.

**Validation:** Kai Zhao, Cuichan Lin, Hon-Cheong So, Zhixiang Lin.

**Visualization:** Kai Zhao.

**Writing – original draft:** Kai Zhao, Hon-Cheong So, Zhixiang Lin.

**Writing – review & editing:** Kai Zhao, Pak Chung Sham, Hon-Cheong So, Zhixiang Lin.

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
