## [Decision Letter · Decision Letter 0]

21 Nov 2023

Dear Dr Lin,

Thank you very much for submitting your Methods entitled 'INSIDER: Interpretable Sparse Matrix Decomposition for RNA Expression Data Analysis' to PLOS Genetics.

The manuscript was fully evaluated at the editorial level and by independent peer reviewers. The reviewers appreciated the attention to an important problem, but raised some concerns about the current manuscript. Based on the reviews, we will not be able to accept this version of the manuscript, but we would be willing to review a much-revised version. We cannot, of course, promise publication at that time.

If you decide to revise the manuscript for further consideration at PLOS Genetics, please aim to resubmit within the next 60 days, unless it will take extra time to address the concerns of the reviewers, in which case we would appreciate an expected resubmission date by email to plosgenetics@plos.org.

We are sorry that we cannot be more positive about your manuscript at this stage. Please do not hesitate to contact us if you have any concerns or questions.

Yours sincerely,

Lin Chen, Ph.D.

Academic Editor

PLOS Genetics

Xiaofeng Zhu

Section Editor

PLOS Genetics

Reviewer's Responses to Questions

**Comments to the Authors:**

Reviewer #1: This article proposed a dimension reduction approach for RNA expression data to decompose variation from different biological variables, which overcomes some limitations of the existing methods and was shown to be useful in the real data analysis. My comments are as follows.

1. It is still not very clear to me how to interpret the metagenes together with the covariates, especially the matrix V. Because it seems that V has nothing to do with the covariates, but the metagenes from INSIDER would be different under the models with different covariates. Also, what is the difference in the interpretation of the metagenes obtained from INSIDER compared to those obtained from PCA or factor analysis?

2. The article mentioned that “INSIDER decomposes variation from different biological variables”, but only the discrete covariates have been discussed. Is INSIDER also able to handle the continuous covariates?

3. In the real data analysis of BrainSpan data, development stages and brain regions are used as covariates. However, the development stages are in chronological order. Does INSIDER handle the ordinal variables and the unordered categorical variables in the same manner?

4. In the section “Metagenes reveal development trajectories”, it said “The expression level of the most variable metagene is high before late fetal stages and decreases to a low level after early childhood”. I wonder how the expression level of the metagene is calculated. Is there an explicit formula of the metagene? Could the author give more details?

5. In the enrichment analyse for a metagene, say the 2nd metagene, it is not clear to me how to conduct the analyse because the metagene is a latent variable. It would be beneficial if more details could be provided.

Reviewer #2: Dimension reduction is critically important in data exploration analysis. Particularly it provides the stepstone to understand the biological signals in high dimensional RNA-seq data. In this paper, the authors propose INSIDER, a general and flexible statistical framework based on matrix factorization. INSIDER decomposes variation from different biological variables and their interactions into a shared low-rank latent space. Particularly, it introduces the elastic net penalty to induce sparsity while considering the grouping effects of genes. It can achieve dimension reduction of high-dimensional data (of >=3 dimensions), as opposed to conventional methods (e.g., PCA/NMF) which generally only handle 2D data (e.g., sample ´ expression). Besides, it enables computing 'adjusted' expression profiles for specific biological variables while controlling variation from other variables.

Overall, the idea is novel and to develop new method to explain the data. better. INSIDER is a general and flexible framework. It can capture interactions between variables or covariate and encourages sparsity in latent representations by imposing the elastic net penalty. The advantages have been highlighted. The paper is well written and the motivation is explained quite clear.

My minor concern is as follows,

1. To use INSIDER well, the users have to know the major variation sources. For example, it’s quite typical for expression data with donors, phenotypes, tissues information. But sometimes those information is not available. How should the method be used?

2. The authors added the phenotype by tissue interactions as an example. To make the method more general, it’s better to having one version without interaction term. It’s also good to demonstrate how the interaction term improves the results.

3. To highlight the high-dimensional data (of >=3 dimensions), it’s better to have more examples to show high-dimensional data is common and quite general case in practice.

4. The choice of K is a multi-scale problem. How should we choose it in practice.

5. It’s better to provide software in Github.

From the editorial board:

Two reviewers and the editorial team have reviewed the manuscript. They showed overall enthusiasm for the proposed method as a general and principled approach to dimension reduction, while also offering several suggestions as follow. Please carefully review those questions and suggestions and respond to them accordingly.

Firstly, the authors claim that the INSIDER method can manage missing data. However, this aspect isn't explored in depth. The effect of significant and complex missing data remains uncertain. It would be beneficial to conduct further evaluations through simulation studies.

Additionally, as one reviewer highlighted, the process for determining the rank K in the latent space is vague. A more clear, justifiable, and efficient method for the choice is needed.

Regarding the analysis of real data, the comparison to standard and frequently used methods like PCA or GLM-based approaches is inadequate. While some discussions are present, more detailed comparative analyses are required.

Lastly, showing additional simulation results in cases of model misspecification would be informative.

**Have all data underlying the figures and results presented in the manuscript been provided?**

Reviewer #1: None

Reviewer #2: Yes

PLOS authors have the option to publish the peer review history of their article (what does this mean?). If published, this will include your full peer review and any attached files.

Reviewer #1: No

Reviewer #2: No

---

## [Decision Letter · Decision Letter 1]

20 Feb 2024

Dear Dr Lin,

We are pleased to inform you that your manuscript entitled "INSIDER: Interpretable Sparse Matrix Decomposition for RNA Expression Data Analysis" has been editorially accepted for publication in PLOS Genetics. Congratulations!

Yours sincerely,

Lin Chen, Ph.D.

Academic Editor

PLOS Genetics

Xiaofeng Zhu

Section Editor

PLOS Genetics

Comments from the reviewers (if applicable):

The reviewers are satisfied with the revision and I would recommend to accept the work.

Reviewer's Responses to Questions

**Comments to the Authors:**

Reviewer #1: The authors have addressed my comments. No more comments.

Reviewer #2: The authors have fully addressed my concerns by performing additional experiments and adding discussions. The manuscript has been greatly improved.

One follow-up suggesting for optimal k selection is as follows. The training and test data split and RMSE calculation is reasonable and rigorous. In dimension reduction, people use the variance explained by the k latent variables to better understand the data structure. It would be nice to provide this index for INSIDER method and also compare with different k choices with shuffled data.

**Have all data underlying the figures and results presented in the manuscript been provided?**

Reviewer #1: None

Reviewer #2: Yes

PLOS authors have the option to publish the peer review history of their article (what does this mean?). If published, this will include your full peer review and any attached files.

Reviewer #1: No

Reviewer #2: No

**Data Deposition**

http://datadryad.org/submit?journalID=pgenetics&manu=PGENETICS-D-23-01170R1

**Press Queries**

---

## [Editor Report · Acceptance letter]

4 Mar 2024

PGENETICS-D-23-01170R1 

INSIDER: Interpretable Sparse Matrix Decomposition for RNA Expression Data Analysis 

Dear Dr Lin, 

We are pleased to inform you that your manuscript entitled "INSIDER: Interpretable Sparse Matrix Decomposition for RNA Expression Data Analysis" has been formally accepted for publication in PLOS Genetics! Your manuscript is now with our production department and you will be notified of the publication date in due course.

With kind regards,

Anita Estes

PLOS Genetics

On behalf of:
